# Evaluation of Photocatalytic Performance of Nano-Sized Sr_0.9_La_0.1_TiO_3_ and Sr_0.25_Ca_0.25_Na_0.25_Pr_0.25_TiO_3_ Ceramic Powders for Water Purification

**DOI:** 10.3390/nano12234193

**Published:** 2022-11-25

**Authors:** Aleksandra Jovanoski Kostić, Nikola Kanas, Vladimir Rajić, Annu Sharma, Subramshu S. Bhattacharya, Stevan Armaković, Maria M. Savanović, Sanja J. Armaković

**Affiliations:** 1University of Novi Sad, Faculty of Sciences, Department of Chemistry, Biochemistry and Environmental Protection, 21000 Novi Sad, Serbia; 2University of Novi Sad, Institute BioSense, 21000 Novi Sad, Serbia; 3University of Belgrade, INS Vinča, Department of Atomic Physics, 11000 Belgrade, Serbia; 4Nanofunctional Materials Technology Centre, Department of MME, IIT Madras, Chennai 600001, India; 5University of Novi Sad, Faculty of Sciences, Department of Physics, 21000 Novi Sad, Serbia; 6Association for the International Development of Academic and Scientific Collaboration (AIDASCO), 21000 Novi Sad, Serbia

**Keywords:** UV degradation, pharmaceuticals, pindolol, application nanomaterials

## Abstract

Water pollution is a significant issue nowadays. Among the many different technologies for water purification, photocatalysis is a very promising and environment-friendly approach. In this study, the photocatalytic activity of Sr_0.9_La_0.1_TiO_3_ (SLTO) and Sr_0.25_Ca_0.25_Na_0.25_Pr_0.25_TiO_3_ (SCNPTO) nano-sized powders were evaluated by degradation of pindolol in water. Pindolol is almost entirely insoluble in water due to its lipophilic properties. The synthesis of the SCNPTO was performed using the reverse co-precipitation method using nitrate precursors, whereas the SLTO was produced by spray pyrolysis (CerPoTech, Trondheim Norway). The phase purity of the synthesized powders was validated by XRD, while HR-SEM revealed particle sizes between 50 and 70 nm. The obtained SLTO and SCNPTO powders were agglomerated but had relatively similar specific surface areas of about 27.6 m^2^ g^−1^ and 34.0 m^2^ g^−1^, respectively. The energy band gaps of the SCNPTO and SLTO were calculated (DFT) to be about 2.69 eV and 3.05 eV, respectively. The photocatalytic performances of the materials were examined by removing the pindolol from the polluted water under simulated solar irradiation (SSI), UV-LED irradiation, and UV irradiation. Ultra-fast liquid chromatography was used to monitor the kinetics of the pindolol degradation with diode array detection (UFLC–DAD). The SLTO removed 68%, 94%, and 100% of the pindolol after 240 min under SSI, UV-LED, and UV irradiation, respectively. A similar but slightly lower photocatalytic activity was obtained with the SCNPTO under identical conditions, resulting in 65%, 84%, and 93% degradation of the pindolol, respectively. Chemical oxygen demand measurements showed high mineralization of the investigated mixtures under UV-LED and UV irradiation.

## 1. Introduction

Due to increased consumption and inappropriate disposal, pharmaceuticals have become one of the most prominent groups of emerging contaminants [1,2]. β-blockers can be detected in water environments because they are one of the most prescribed pharmaceuticals [1,2]. While several countries pay considerable attention to the presence of β-blockers in water, others are less aware and do not have legislative measures and regulations for the environmental monitoring of these contaminants [3]. After digestion in the human digestive tract, β-blockers can be excreted unchanged or undergo specific metabolic changes in the water environment. Interactions between β-blockers and other coexisting compounds can produce an additive or synergistic effect, representing an additional environmental hazard. This results in interference or disorder in the physiological properties of aquatic organisms, and consequently the disruption of the aquatic ecosystem and of human health [4,5,6]. Pindolol (1-(indol-4-yioxy)-3-isopropylaminopropan-2-ol) is employed in treating hypertension, angina, glaucoma, and pregnancy, and nowadays it is also used as an antidepressant. Due to its lipophilic properties, pindolol is almost entirely insoluble in water [7,8,9].

The presence of pharmaceuticals in aquatic systems is significant for this study due to their effect on aquatic flora and fauna. The impact of pharmaceuticals on organisms in environmental waters manifests itself relatively slowly due to their usually slow accumulation. Moreover, their harmful effects cannot be observed until a worrying level is reached, after which the process is irreversible. Concerning the influence of pharmaceuticals on aquatic organisms, the problem is not only their stability; it is also often the case that the parent compounds are converted back to the starting compound, which is why the harmful effect is multiplied [10]. Hence, it is essential to develop efficient methods to separate them from wastewater since they may express potential health risks.

Advanced oxidation processes (AOPs) can remove β-blockers with an efficiency of up to 90%, especially when combined with photocatalytically active materials [11]. Du et al. [12] have pointed out that it is essential to rationally design nanostructures with the potential to provide desirable multi-functionalities. Moreover, the same authors emphasized that the photocatalytic activities of Ce_0.90_Sn_0.10_O_2-δ_ remain consistent in four cyclic cycles, indicating that CeO_2-δ_ doped with Sn possesses excellent strength for application.

Perovskite-type catalysts (e.g., ABO_3_) are attractive options among numerous materials used due to their unique structural features. A high tolerance factor enables co-doping and co-substitution at A- and B-sites while maintaining the same crystal structure. This is beneficial for tuning the various performances of these materials.

Perovskites have been known to exhibit excellent photocatalytic activity, and hence a more in-depth analysis of these materials would be beneficial. The photocatalytic activities of SrTiO_3_ and BaTiO_3_, including their oxides, on the decomposition of water have been reported [13]. Perovskite is considered the ideal structure for the assembly of photocatalysts [14]. However, SrTiO_3_ displays even greater photocatalytic activity than TiO_2_ due to the long lifetime of its electron-hole pair recombination, intense catalytic activity, high thermal stability, chemical stability, and good biological compatibility [15,16,17]. What limits the practicality of SrTiO_3_ is the wide bandgap energy. However, it only absorbs in the UV region, which constitutes 5% of solar irradiation. This imposes a limitation on its practical use. Numerous strategies have been employed to enhance the photocatalytic activity of SrTiO_3_ by doping with non-metal [18] and metal [19] ions. It has been noticed that among the candidates for the modification of SrTiO_3_ materials, La has stood out. Numerous findings have shown the advantages of La-doped SrTiO_3_. Doping with La^3+^ increases surface area and adsorption capacity, resulting in the enhancement of the photocatalytic performance of SrTiO_3_ in the photocatalytic degradation of pollutants [20]. Park et al. provided a practical method for surface doping by La ions on SrTiO_3_ nanotubes which did not find practical application [21]. Yang et al. found that under UV radiation a 5 mol.% La-doped SrTiO_3_ photocatalyst reduced Cr(VI) by 84%, whereas the undoped sample reduced Cr(VI) by only 54% [20]. La-doped SrTiO_3_ nanotubes degraded 50% of 2-naphthol under LED radiation after 360 min [16]. Thus far, multifunctional materials such as Sr_0.9_La_0.1_TiO_3_ (SLTO) and Sr_0.25_Ca_0.25_Na_0.25_Pr_0.25_TiO_3_ (SCNPTO) and their photocatalytic activities have not been studied. However, it can be expected that, as perovskite-type oxides, these compositions or their composites could exhibit good photocatalytic activity. SCNPTO possesses configurational entropy obtained by a careful A-site substitution of three cations with different radii, maintaining the charge neutrality of pristine SrTiO_3_. On the other hand, n-type electrical conductivity is dominant when introducing La to SrTiO_3_, and less configurational entropy is present in compassion with SCNPTO. Taking the advantages of doped materials into account, we modified the starting materials and synthesis procedure to prepare La-doped SrTiO_3_. Considering the data in the literature suggesting that Ca [22], Na [23], and Pr [24] are the most promising candidates for the modification of SrTiO_3_ materials, these elements, in addition to La, were used to modify SrTiO_3_. A discussion of the differences between SLTO and SCNPTO is included, and essential insights concerning the photocatalytic performance of the system are provided.

The as-prepared photocatalysts were characterized for their phase composition, morphology, and chemical composition using X-ray powder diffraction (XRD), high-resolution scanning electron microscopy (HR-SEM), energy-dispersive X-ray spectroscopy (EDX), particle size analysis (PSA), and nitrogen adsorption analysis using the Brunauer–Emmett–Teller (BET) method. The photocatalytic activities of the synthesized samples were investigated via the degradation of pindolol under simulated solar irradiation (SSI), UV-LED irradiation, and UV irradiation. A chemical oxygen demand (COD) test was used to determine the amount of oxygen necessary to oxidize organic matter in samples in which photocatalysis has taken place for 240 min. The studied materials were highly efficient in removing the parent compounds and intermediates formed during the degradation. Since the pindolol turned out to be highly stable in the aquatic environment, the aim was to find a more efficient way for its removal using the newly synthesized perovskites.

## 2. Experimental and Computational Details

### 2.1. Chemicals and Solutions

All chemicals were used in their as-received form, without further purification: 85% H_3_PO_4_ (Lachema, Neratovice Czech Republic), acetonitrile (J.T. Baker), K_2_Cr_2_O_7_ (Sigma–Aldrich, Taufkirchen Germany), H_2_SO_4_ (95–97%, Pharma Hemija, Šabac Serbia), HgSO_4_ (≥99.5%, Sigma–Aldrich, Taufkirchen Germany), Ag_2_SO_4_ (≥99.9, Sigma–Aldrich, Taufkirchen Germany), and HOOCC_6_H_4_COOK (≥ 99.5%, Merck, Taufkirchen Germany). The active component of β-blockers, pindolol (Sigma–Aldrich, Taufkirchen Germany), was used as received (≥99% purity), and all solutions were made using ultrapure water (UPW).

### 2.2. Materials Synthesis

SCNPTO perovskite with a configurational entropy was prepared by the reverse co-precipitation (RCP) method with various substitutions of Ca, Na, and Pr for Sr at the A-site, whereas Ti was fixed at the B-site. The powder was synthesized using nitrate precursors for strontium (99%, Sr(NO_3_)_2_, Alfa Aesar, Kandel Germany), calcium (99% Ca(NO_3_)_2 ·_ 4H_2_O, Alfa Aesar, Heysham UK), sodium (99%, NaNO_3_, Alfa Aesar, Kandel Germany), praseodymium (99.9%, Pr(NO_3_)_3_·6H_2_O, Alfa Aesar, Kandel Germany), and an organometallic precursor, titanium n-butoxide (99% liquid Ti[O(CH_2_)_3_CH_3_]_4_, Alfa Aesar, Kandel Germany) for titanium. The composition was prepared to a concentration of 1M. For the system with four elements at the A-site, the nitrate precursors were taken in proportion to obtain a final composition of 0.25 molar for each cation, whereas the Ti[O(CH_2_)_3_CH_3_]_4_ was taken so as to obtain a 1 mol dm^−3^ concentration of Ti for the B-site. The nitrate precursors were dissolved in de-ionized water (DI). The Ti[O(CH_2_)_3_CH_3_]_4_ was dissolved in 10 mL of ethylene glycol for all the systems. The nitrate and Ti precursor solutions were mixed together using a magnetic stirrer until a clear solution was obtained. The final precursor was then precipitated in liquid ammonia solution (Merck, Darmstadt Germany) and maintained at a pH of 11 in a controlled drop-wise manner in an ultrasonic bath. The precipitates were filtered and dried overnight in a hot air oven. The powders were then calcinated in a conventional furnace at 700 °C for 5 h. SLTO, on the other hand, was produced by spray pyrolysis (CerPoTech, Trondheim Norway).

### 2.3. Characterization Methods

In order to validate the phase purity of the SCNPTO and SLTO, X-ray diffraction patterns were recorded in the 2θ range of 20−75° using Cu-Kα radiation with a scanning rate of 0.5°/min (Rigaku MiniFlex 600 diffractometer). The particle size and the morphology were determined by high-resolution scanning electron microscopy, HR-SEM (FEI SCIOS 2 Dual Beam microscope), with an EDS detector. EDS was performed using an INCAx-sight detector and an ‘‘INAx-stream’’ pulse processor (Oxford Instruments), Appendix A. The specific surface area of the powders was determined from nitrogen adsorption studies (NOVAtouch, Quantachrome Instruments) using the BET method, while the particle size distributions were obtained with a PSA (Litesizer 500, Anton Paar) using ultrasonicated, low-concentrated suspensions (~3 wt.%). For the determination of the band gap energies (E_g_) of the two materials, diffuse reflectance measurements were performed using a UV–VIS spectrophotometer, Evolution 600 (Thermo Scientific), with a DRA-EV-600 diffuse reflectance integrating sphere accessory in the range between 240 nm and 840 nm with a 1 nm step and a speed of 10 nm/min.

### 2.4. Photocatalytic Activity

The photocatalytic activities of the powders were evaluated from the degradation of a solution of pindolol. Experiments were performed using 20 cm^3^ of 0.05 mmol dm^−3^ pindolol containing 1.0 mg cm^−3^ catalyst. The results of the photocatalytic efficiency were compared with the results of the direct photolysis. All experiments were performed at the natural pH (ca. 9.50). The solution was placed in an ultrasonic bath for 10 min to keep the catalyst particle size uniform and achieve adsorption equilibrium. The vessel containing the suspension was placed on a magnetic stirrer and maintained at 25 ± 0.5 °C with stirring in a stream of O_2_ for 5 min before irradiation. In each case, during irradiation, the solution was continuously stirred on a magnetic stirrer, and a flow of O_2_ (3.0 cm^3^ min^−1^) was continued, thus achieving its constant concentration. The SSI was performed using a 50 W halogen lamp (Philips) with an intensity of 0.1 W cm^−2^ in the visible region and 2.2 × 10^−4^ W cm^−2^ in the UV region, and a 5W UV-LED Lamp was used as a source of LED radiation (Enjoydeal, China, type: MR16 AC 85-265V/12). As a source of artificial UV radiation, a high-pressure mercury lamp (Philips, HPL-N, 125 W with emission strips in the range of UV radiation at 304 nm, 314 nm, 335 nm, and 366 nm, and an emission maximum at 366 nm) with a corresponding concave mirror was used.

More details about the degradation procedures is given in the Appendix A.

### 2.5. Analytical Procedures

Ultra-fast liquid chromatography with diode array detection (UFLC–DAD, Shimadzu) was used to monitor the kinetics of the pindolol degradation. Aliquots of 0.30 cm^3^ were taken from the reaction mixture at the beginning of the experiment and at regular time intervals. Aliquot sampling caused a maximum volume variation of ca. 10% in the reaction mixture. More details about the UFLC–DAD procedures are provided in the Appendix A.

The mineralization was determined according to the United States Environmental Protection Agency standard method 410.4. The COD concentration was determined spectrophotometrically by measuring the absorbance of the formed Cr^3+^ according to the calibration curve (Appendix A). Aliquots of 2.5 cm^3^ samples were taken, and the COD measurements were performed according to the manual method. More details about the COD measuring are provided in the Appendix A.

### 2.6. Computational Methods

To understand the photocatalytic degradation process, we performed a detailed computational analysis based on density functional theory (DFT) calculations. To learn the essential reactive properties of the pindolol molecule and the band structures of the applied perovskites, analysis by molecular and periodic DFT calculations was carried out.

Before molecular DFT calculations were performed, the MacroModel [25] program was used with the OPLS3e force field [26,27,28,29] to generate and optimize all possible conformers. All generated conformations were optimized at the DFT level with a B3LYP [30] functional and 6–31 G(d,p) [31,32,33] basis set. Of all conformers, the ten lowest energy conformers were re-optimized with tighter SCF conditions. Finally, we selected the conformer with the lowest energy for further molecular DFT calculations. Frequency calculations at the same level of theory were performed to confirm that the true ground states has been identified. The lowest energy conformer of the pindolol was then subjected to calculations of p*K*_a_ constants, employing the Epik program and the empirical method [34,35,36]. The Jaguar [37,38] program, which is incorporated in the Schrödinger Materials Science Suite (SMSS) 2022-1 (as is the MacroModel program), was used for the molecular DFT calculations.

To understand the electronic structures of the applied photocatalysts, their crystal structures were subjected to geometrical optimizations and band structure calculations by applying periodic DFT computations. The periodic DFT calculations were performed using the LCAO engine implemented in the QuantumATK [39,40,41,42] modeling package. A Perdew–Burk–Erzhenorf (PBE) [43] functional was applied along with the pseudopotentials [44] and double zeta polarized basis set for optimizations, while pseudo Dojo pseudopotentials [45] and a medium basis set were used for band structure calculations. Geometrical optimizations were performed until the forces were lower than 0.05 eV/Å, while the maximum step size was 0.1 Å. K-point sampling was set to 2 × 2 × 2 for optimizations, while for the band structure calculations, it was set to 4 × 4 × 4. The broadening was set to 25 meV. Fermi–Dirac smearing was used, while the density mesh cut-off was set to 130 Hartrees.

## 3. Results and Discussion

### 3.1. Powder Characteristics

X-ray diffractograms of the two ceramic powders are shown in Figure 1. The peaks were sharp, with high intensities, indicating a high degree of crystallinity. The diffraction lines of the phase-pure SCNPTO were observed, whereas the SLTO (CerPoTech) had the same phase-pure pattern, but combined with minor low-intensity diffraction lines of SrCO_3_ and TiO_2_. This was probably a result of long-term exposure to an ambient condition.

HR-SEM micrographs of the ceramic powders are displayed in Figure 2. The particle sizes of the SCNPTO seem to be 60–70 nm, while the SLTO particles are even smaller due to the spray pyrolysis synthesis technique. However, both powders possess spherical morphology with a certain degree of soft agglomeration.

Crystallite sizes (CSs) were obtained as estimated integral values for each peak position using HighScore Plus (Philips’ program). Average CSs of about 27 nm and 33 nm were calculated for the SLTO and SCNPTO, respectively, and the data are graphically presented in Appendix A. It can be observed that CS possess slightly smaller values than particles size, as expected [46].

Since smaller particles provide a larger surface area and have a larger surface free energy, the SLTO particles were expected to demonstrate a more pronounced tendency toward soft agglomeration. Figure 3 shows the relative fraction (%) as a function of particle size diameter. The particle size distribution (volume) was challenging since the detected signals mainly originate from agglomerates. Hence, according to the PSA, the most frequent size of the SCNPTO and SLTO agglomerates was 1 µm and 3 µm, respectively, even though much smaller agglomerates could still be observed from the SEM micrographs.

The peaks at about 200 nm in both cases indicate a narrow size distribution of the less agglomerated particles in the powders, arising from the breakage of the soft agglomerates during ultrasonication prior to the measurements. The higher intensity peak of the SCNPTO at 200 nm indicates a larger fraction of less agglomerated particles compared with the SLTO powder. A good correlation between Figure 2 and Figure 3 is evident and confirms that the SLTO powder possess a higher degree of soft agglomeration, which results in a lower surface area. Additionally, this is supported by the BET results, where specific surface areas of 27.6 m^2^ g^−1^ and 34.0 m^2^ g^−1^ for the SLTO and SCNPTO powders were determined, respectively.

### 3.2. Degradation Efficiency

Direct photolysis with SSI didn’t prove to be efficient since only 8% of the pindolol was degraded/removed after 240 min. More efficient degradation was observed under the influence of UV-LED and UV irradiation, wherein pindolol degradation of 12% and 36%, respectively, was obtained for the same time (Appendix A). These results were expected, considering that the absorption maximum of pindolol is at 217 nm. Irradiation under the shorter wavelengths of the UV lamp contributed to the higher efficiency of degradation compared with the SSI and UV-LED lamps (Figure 4).

Comparing the photocatalytic activities of the SLTO and SCNPTO with that of direct photolysis (Appendix A and Figure 4), it can be seen that, in the case of both perovskites, the degradation of the pindolol was considerably faster for all three irradiation sources. After 240 min of irradiation, while using the SLTO, 68%, 94%, and 100% of the pindolol was removed under the influence of SSI, UV-LED, and UV irradiation, respectively. Somewhat lower degradation efficiencies were observed when the SCNPTO was used under identical conditions and irradiation sources. In these cases, 65%, 84%, and 93% of the pindolol was removed under SSI, UV-LED, and UV irradiation, respectively.

The results suggest that the efficiency of photocatalytic degradation varies depending on the photocatalyst used. An increasing number of research groups are working on elucidating the mechanisms of these complex processes, which shows the importance of this topic [47,48,49]. Yang et al. examined the kinetics and proposed a general mechanism for the photocatalytic degradation of three β-blockers (atenolol, metoprolol, and propranolol). The reaction with ^•^OH was the most significant in degrading the mentioned pharmaceuticals [47]. Romero et al. also investigated the photocatalytic degradation intermediates of β-blockers. They found that the main pathway of photocatalytic degradation was via hydroxylation of the aromatic ring or side chain. In addition, there was a shortening of the aliphatic parts of the molecule [48]. Results from Armaković et al. illustrate that the reactive radicals play a significant role in the degradation of pindolol compared with degradation using h^+^ [49].

The faster degradation of pindolol by the SLTO, which has a lower surface area than the SCNPTO, is an interesting result that was further addressed through various experimental and computational approaches.

From the computational standpoint, we aimed to calculate and compare the band structures of the applied materials. However, before performing the band structure calculations, we had to generate suitable models of the applied materials and perform geometrical optimizations. The structures of the SLTO and SCNPTO were modeled by taking the 2 × 2 × 2 supercell of pristine SrTiO_3_ and by making suitable modifications to that supercell. The considered 2 × 2 × 2 supercell of pristine SrTiO_3_ consisted of 40 atoms: 8 atoms of Sr, 8 atoms of Ti, and 24 atoms of O. In the case of the SLTO, 10% of the Sr atoms were replaced with La atoms. Modeling the exact SLTO structure would require a colossal supercell with more than 200 atoms, which would be computationally too demanding for DFT calculations. This imposed the necessity to take into account the approximated model. Since our considered supercell contains eight Sr atoms, replacing one Sr atom with one La atom corresponds to the formula Sr_0.88_La_0.12_TiO_3_, which is more than a reasonable approximation of the formula Sr_0.9_La_0.1_TiO_3_. The geometrically optimized structures of the SLTO and SCNPTO are presented in Appendix A.

In both cases, the optimization was performed using the condition that the type of lattice (Bravais lattice) was preserved, which in the present case was a simple cubic lattice, as indicated by the experimental results. The structures obtained after the geometrical optimizations were subjected to band structure analysis. The calculated band structures are presented in Figure 5.

Ultraviolet–visible spectroscopy (UV–VIS) was conducted on both powders to compare the experimental and computational results. Due to the high reflection compared with the diffusion scattering created by the SLTO, the data could not be fitted appropriately. Therefore, the obtained band gap of about 3.50 eV possesses a pronounced uncertainty.

On the other hand, the experimentally obtained band gap for the SCNPTO was 3.39 eV. Here we could observe similar differences between the experimental and calculated values for the SLTO (0.45 eV) and the SCNPTO (0.70 eV), respectively (Appendix A). The observed discrepancy was as expected and matches with a trend in the literature.

The results presented in Figure 5 explain the efficiency of the photocatalytic degradation of the pindolol using the studied materials. In the first place, the experimental results show that photocatalysis under SSI was the least efficient. This was expected since SrTiO_3_ belongs to a group of wide-range band gap materials, because of which photons of higher energy are required to promote electrons from the valence band (VB) to the conduction band (CB). The band structures calculated for the materials used in this study indicate that the energy difference between the VB and CB is also wide, as shown in Figure 5a,b, explaining why both materials turned out to be photocatalytically active under UV irradiation.

However, the efficiency of the La-containing system was considerably higher, which can be explained by a detailed analysis of the calculated band structures. In both cases, the Fermi level entered the CB, indicating that the materials are conductors and that the electrons are the charge carriers. The Fermi level entered just slightly into the CB, so the charge carriers’ density was lower. The conducting properties were still principally regulated by the promotion of electrons from the VB to the CB. When the band structure was enlarged in the vicinity of the Fermi level, as shown in Figure 5c,d, it can be seen that the densities of the bands below the Fermi level were much higher in the case of the La-doped material. This led to a higher density of charge carriers and better photocatalytic properties as the electrons were promoted from the VB to the CB via UV irradiation.

The calculated band structures also indicate another reason why the La-doped material possesses better photocatalytic properties. The calculations show that the energy difference between the VB and CB in the case of the SCNPTO was 2.69 eV, while that in the case of the SLTO was 3.05 eV. Recently it has been reported that the presence of Pr ions tends to lower the band gap in high-entropy oxides [50]. The much higher band gap in the case of the La-doped material is beneficial in preventing the recombination of electrons and holes to a greater extent than the SCNPTO, leading to a more efficient photocatalytic degradation.

It is also worth mentioning that the experimental results confirmed the computational results in one more aspect. In terms of photocatalysis, the fact that the electrons were the charge carriers indicates that the oxidative reactive species are responsible for the degradation of pindolol. This is confirmed by the variation in pH values since it has been seen that this parameter changes (Figure 6) when the electrons interact with water to produce oxidative reactive species.

Monitoring the pH value of the solution during the photocatalytic process provided insight into the global changes in the studied system (Figure 6), although changes in pH value directly correspond to the kinetics of degradation only in cases of much simpler molecules than pindolol [51]. Additionally, the pH value of the solution can significantly influence the electrostatic interactions between the surface of the photocatalyst and substrate, which was the motivation for us to monitor changes in pH value. The starting pH was around 9.40, and during degradation this value changed by about ± 1.0.

As shown in Figure 6, during degradation, the pH value initially decreased in all cases, probably due to the formation of acid intermediates. A small exception was observed in the case of the SCNPTO under SSI irradiation. The slight increase in pH value after approximately 30 min under UV-LED irradiation might have been the consequence of the formation of NH_4_^+^ ions and consumption of H^+^ ions.

The similar trend in the changes in the pH values during the degradation of pindolol indicates potentially similar mechanisms. The pH value of the suspension being lowest after 240 min of irradiation with a UV lamp agrees well with the observed degree of mineralization (Table 1). Namely, the high degree of mineralization of around 80% in both cases indicates that most intermediates had been removed from the water solution. This led to the resultant pH value being close to the natural pH value of the pindolol solution (ca. 7.50).

The pH effects governing the degradation efficiency include water dissociation, which influences the level of ^•^OH radicals, and the oxidative power of photogenerated h^+^. However, the role of these effects cannot be readily estimated because the influence of h^+^ is favored in acidic conditions. At the same time, ^•^OH radicals dominate in neutral and alkaline conditions. Since the pH value of the solution was above 8.00 in all cases, it can be concluded that the ^•^OH radicals were the most important for the high level of degradation of the pindolol and its intermediates. This conclusion fully conforms with the computational results, which will be presented later.

The changes in pH value observed through our experiments motivated us to study the p*K*_a_ values of pindolol. Specifically, through DFT calculations and a protocol implemented into the Jaguar program of the SMSS package, we calculated the p*K*_a_ value of the pindolol molecule. The 3D structure of the identified lowest energy conformer of pindolol is presented in Figure 7.

The optimized structure presented in Figure 7 was used to calculate the p*K*_a_ value using an empirical approach. The experimental p*K*_a_ value of pindolol was 9.49 [52]. The calculations gave a result of 9.39, which shows excellent agreement between experiment and theory. However, in terms of photocatalytic degradation, it is even more critical to identify the molecular site with the lowest p*K*_a_ value because this might be where degradation starts. According to both methods, the lowest p*K*_a_ value in the case of pindolol was calculated for the hydrogen atom H26, connected to the nitrogen atom.

Pindolol contains an amino group that is protonated at pH < p*K*_a_, as follows (reaction 1):
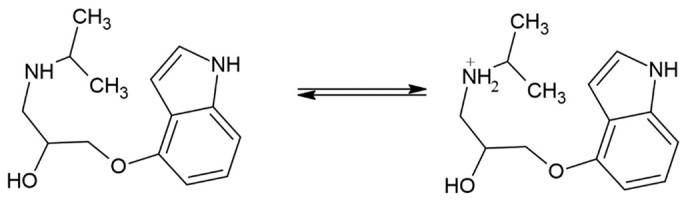
(1)

The p*K*_a_ value of pindolol is about 9.5 and the isoelectric point of perovskite is in the range of about 6.0 < pH < 10.0 [53,54,55], and this favors electrostatic attraction between the negatively charged surface of the material and the protonated pindolol, which explains the extremely high decomposition efficiency at the mentioned pH values (Figure 6).

It has been well established that the degradation of organic compounds is frequently followed by the generation of intermediates that are potentially more toxic to the environment than the starting compound. Hence, it becomes essential to study the degree of mineralization during degradation. The COD measurements (Table 1) show that even after the complete removal of pindolol in the presence of SLTO (Figure 4) under UV irradiation for 240 min, around 19% of the organic compounds (measured as COD) remained in the system. A high degree of mineralization was also observed for the SCNPTO perovskite. On the other hand, the degree of mineralization under UV-LED irradiation for 240 min was about 50% in both cases.

If we compare the pindolol removal efficiency of the tested materials with our earlier research in the literature [49], we can conclude that under the same degradation conditions, the degradation efficiency is similar in the presence of commercial ZnO and TiO_2_ (Hombikat and Degussa). Still, the degradation efficiency is significantly lower in the presence of commercial TiO_2_ (Wackherr) [49]. However, the most significant of the obtained results lies in the efficient degradation of the intermediates formed during the process and the non-selectivity of the investigated perovskites. Specifically, the mineralization measured as total organic carbon ranged between 20% and 40% under the effect of UV radiation [49], while our results showed a degree of mineralization higher than 70% for the same analysis time. One explanation for the high degree of mineralization efficiency may be the presence of defects and oxygen vacancies that were tailored by doping with the metal cations [56].

All in all, both the SLTO and SCNPTO demonstrated very promising photocatalytic performances in terms of water purification.

## 4. Conclusions

The nano-sized Sr_0.25_Ca_0.25_Na_0.25_Pr_0.25_TiO_3_ (SCNPTO) powder was synthesized by the reverse co-precipitation method using nitrate precursors, whereas the Sr_0.9_La_0.1_TiO_3_ (SLTO) nanopowder was produced by spray pyrolysis (CerPoTech). X-ray diffractograms indicated highly crystallite materials in both cases. HR-SEM micrographs showed the particle size of the SCNPTO to be 50–70 nm, while the SLTO_3_ particles were smaller due to the spray pyrolysis synthesis technique. The most frequent agglomerate sizes of the SCNPTO and SLTO were 1 µm and 3 µm, respectively. BET measurements gave specific surface areas of 27.6 m^2^ g^−1^ and 34.0 m^2^ g^−1^ for the SLTO and SCNPTO powders, respectively. The degradation of the pindolol by direct photolysis with SSI wasn’t efficient, while in the presence of UV-LED and UV irradiation, a high extent of degradation was achieved due to the absorption maximum of pindolol lying in the UV range (217 nm). After 240 min under SSI, UV-LED, and UV irradiation, the SLTO removed 68%, 94%, and 100% of the pindolol, while the SCNPTO removed 65%, 84%, and 93% of the pindolol, respectively. The energy band gaps of the SCNPTO and SLTO, calculated by DFT, were 2.69 eV and 3.05 eV, respectively. The much higher band gap in the case of the La-doped material was beneficial in preventing the recombination of the generated electrons and holes, resulting in better photocatalytic properties. Monitoring the pH value gave insight into the high level of degradation of the pindolol and its degradation intermediates. Since the pH value of the solution was above 8.00 in all cases, and because ^•^OH radicals have a dominant role in neutral and alkaline conditions, it was concluded that the ^•^OH radicals were the most important in the degradation of the pindolol. Through DFT calculations, a p*K*_a_ value of 9.39 was obtained for the pindolol molecule. The lowest p*K*_a_ value in the case of pindolol was obtained for the hydrogen atom H26, connected to the nitrogen atom, which may be the molecular site where degradation starts. The COD measurements showed high mineralization of the pindolol and its degradation intermediates under UV-LED and UV irradiation.

## Figures and Tables

**Figure 1 nanomaterials-12-04193-f001:**
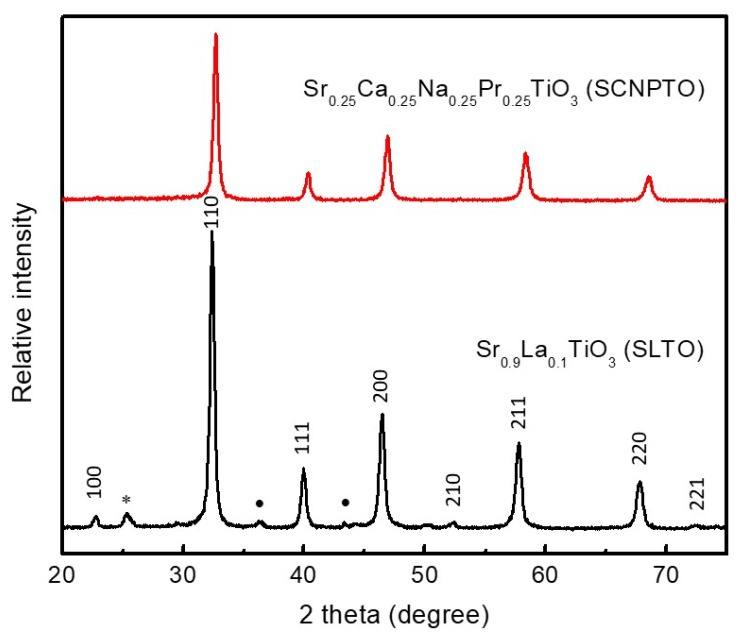
Powder X-ray diffraction patterns of the investigated materials. The main phase Sr_0.9_La_0.1_TiO_3_ with Miller indices is indicated and the PDF card: 04-002-1010 is given as a reference. Minor peaks of SrCO_3_ (PDF card: 05-0418) and TiO_2_ (PDF card: 21-1276) phases are indicated in the SLTO powder (CerPoTech) as (*) and (•), respectively.

**Figure 2 nanomaterials-12-04193-f002:**
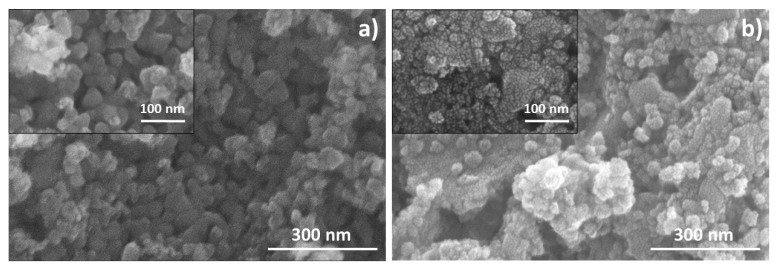
HR-SEM micrographs of (**a**) SCNPTO and (**b**) SLTO powders.

**Figure 3 nanomaterials-12-04193-f003:**
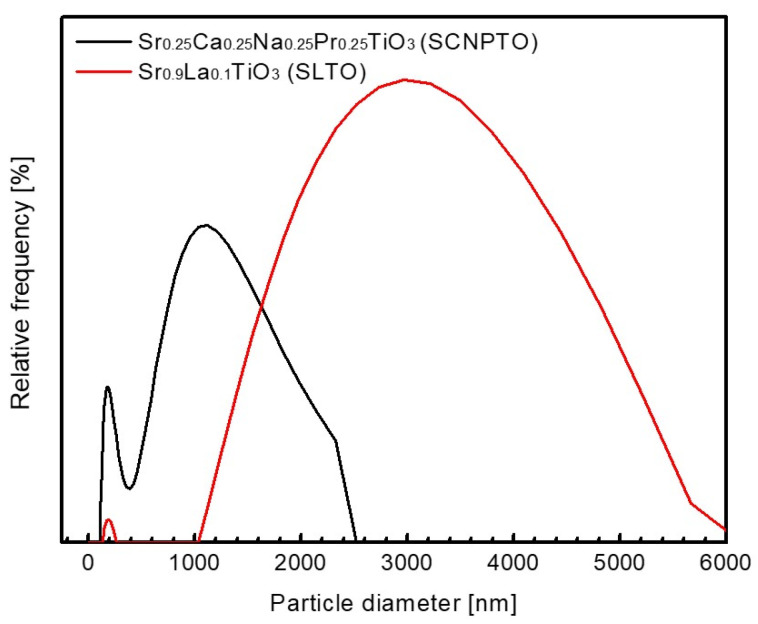
Particle size distribution of investigated materials.

**Figure 4 nanomaterials-12-04193-f004:**
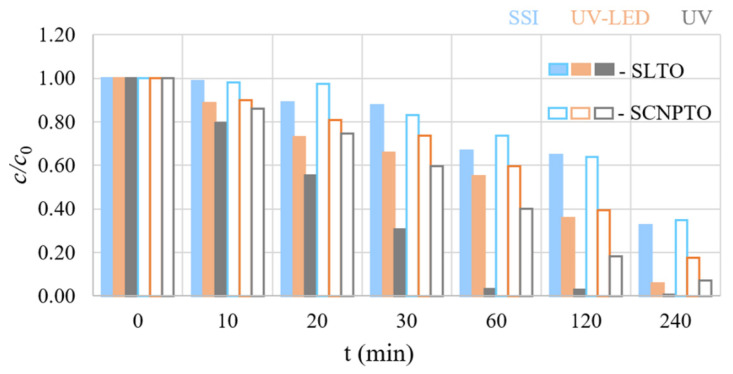
Kinetics of pindolol photocatalytic degradation under different types of irradiation in the presence of perovskites.

**Figure 5 nanomaterials-12-04193-f005:**
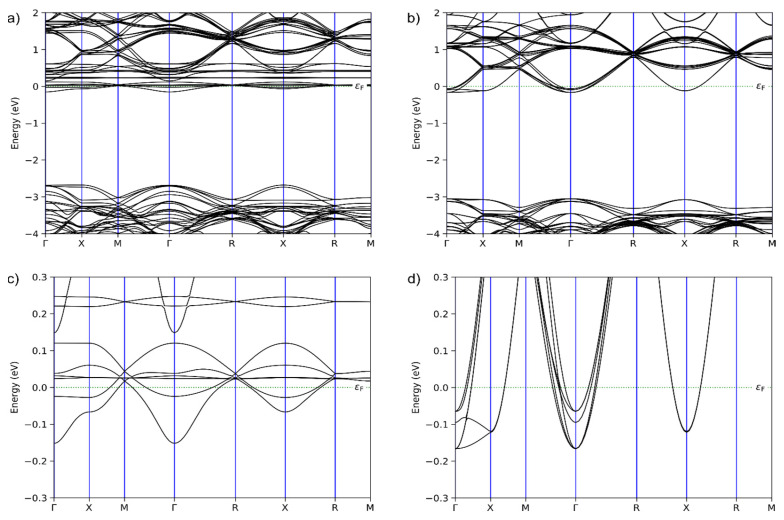
Band structures of (**a**) SCNPTO and (**b**) SLTO and enlarged band structures around the Fermi level of (**c**) SCNPTO and (**d**) SLTO.

**Figure 6 nanomaterials-12-04193-f006:**
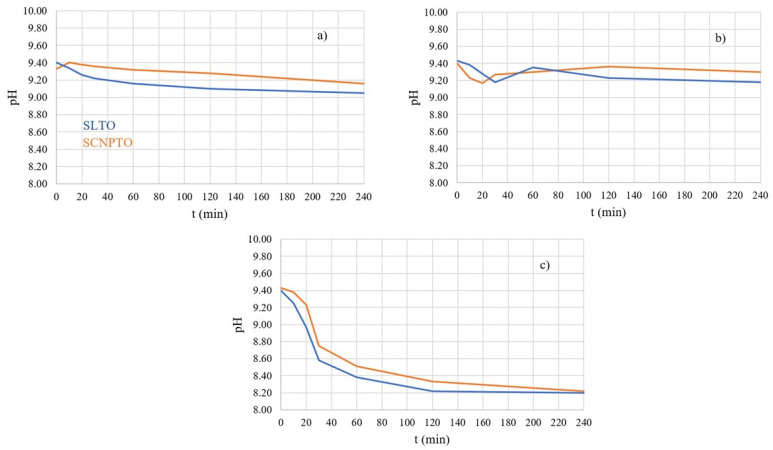
pH change during pindolol photodegradation in the presence of perovskites under (**a**) SSI, (**b**) UV-LED, and (**c**) UV irradiation.

**Figure 7 nanomaterials-12-04193-f007:**
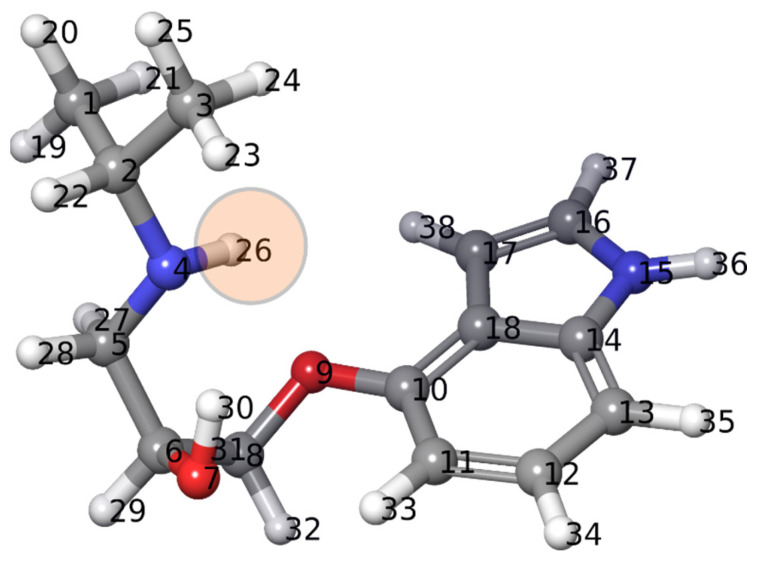
Optimized structure of pindolol used for the calculation of p*K*_a_ value.

**Table 1 nanomaterials-12-04193-t001:** Mineralization of the investigated solutions for different processes of degradation.

Suspension	COD (mg O_2_ dm^−3^)	Mineralization (%)
SSI		
SLTO	73.79	22.5
SCNPTO	79.13	16.9
UV-LED		
SLTO	48.81	48.7
SCNPTO	54.16	43.1
UV		
SLTO	18.48	80.6
SCNPTO	25.61	73.1

COD for PIN suspension was 95.20 mg O_2_ dm^−^^3^.

## Data Availability

Not applicable.

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
