# Peer review of "Evaluation of Photocatalytic Performance of Nano-Sized Sr0.9La0.1TiO3 and Sr0.25Ca0.25Na0.25Pr0.25TiO3 Ceramic Powders for Water Purification"

_nanomaterials, 2022, doi:10.3390/nano12234193_

Round 1

Reviewer 1 Report

The manuscript is dedicated to the synthesis of complex perovskite-type oxides and their application as the photocatalyts for degradation of pinodol in water. Variety of the instrumental techniques was used to characterize both the prepared photocatalytic materials and kinetics and products of photocatalytic process.
Despite the comprehensive characterization, several questions are arised. First of all, the chemical composition of prepared perovskite-type oxides is not discussed. One of the qualitative techniques (EDX, ICP MS, etc.) should be applied to prove stoichiometry of the prepared phases. Secondly, preparation procedure of SLTO should be described. If this compound was purchased from the outside supplier, articular number or etc. should be stated. Thirdly, results of the geometrical optimization of the structures of complex perovskite-type oxides and pinodol should be confirmed experimentally. In particular, the band gap of semiconductors may be determined experimentally using optical spectroscopy. These values may be more relevant then the calculation from “geometrical optimization”.

The following minor issues should be also corrected/commented:
1.    Figure1 –reference to the PDF card for SCNTPO should be added.
2.    Line85 and below – abbreviations PSD/PSA should be defined.
3.    Line212 and Figure 3 – the mean diameter of the small particles equals to 200nm but not 20nm?
4.    Figure4 – bars and lines for the different phases are hard to compare with the each other.
5.    Wavelength characteristics of the used light sources should be moved to the main text because spectral issues of photocatalytic process are discussed.
6.    As stated in section 2.4 - photocatalytic activity was measured at the pH 7,5. However, pH changes in Figure 6 starts from the pH values about 9,5.
7.    Figure 5d belongs to the SLTO phase?

To sum up, the manuscript may be subjected to publication in Nanomaterials after the careful revision according to the points listed above.

Author Response

Journal: Nanomaterials (ISSN 2079-4991)

Manuscript ID: nanomaterials-2017918

Title: Evaluation of photocatalytic performance of Sr0.9La0.1TiO3 and Sr0.25Ca0.25Na0.25Pr0.25TiO3 nano-sized ceramic powders for water purification

Section: Energy and Catalysis

Special Issue: Nanoscale Materials for Water Purification and Catalysis

The authors thank the reviewer for their constructive comments and recommendations. We have taken the comments on board to improve and clarify the Manuscript. Please find below a detailed point-by-point response to all comments.

The reviewers’ comments are typed in black. The responses to reviewers are tagged blue. The changes in the text of the Manuscript are typed in red. The line numbers in responses to reviewers refer to the corrected text.

Response Reviewer 1:

The manuscript is dedicated to the synthesis of complex perovskite-type oxides and their application as the photocatalyts for degradation of pinodol in water. Variety of the instrumental techniques was used to characterize both the prepared photocatalytic materials and kinetics and products of photocatalytic process.

Despite the comprehensive characterization, several questions are arised.

First of all, the chemical composition of prepared perovskite-type oxides is not discussed. One of the qualitative techniques (EDX, ICP MS, etc.) should be applied to prove stoichiometry of the prepared phases.

Thank you very much for this point. EDX analysis was performed on both powders, and the results are now included in Supplementary materials as Table S1. Since EDX provides up to 5 % error on compositional analysis, it is confirmed that both compounds have correct stoichiometries and formulae, considering the ratio between the cations.

We have added information about EDX analysis to the manuscript in 1. Introduction (Line 110) and 2.3. Characterization methods information (Lines 153 and 154), and Table S1 in Supplementary materials (Line 471).

Secondly, preparation procedure of SLTO should be described. If this compound was purchased from the outside supplier, articular number or etc. should be stated.

The article number is unfortunately unavailable. However, polycrystalline SLTO was synthesized by spray pyrolysis of aqueous nitric solutions, standardized by CerPoTech AS (Trondheim, Norway). The synthesis procedure is the same as for all the oxide ceramic powders that CerPoTech produces (https://www.cerpotech.com/products/product-portfolio). We added this information in the Supplementary materials (Materials synthesis).

Thirdly, results of the geometrical optimization of the structures of complex perovskite-type oxides and pinodol should be confirmed experimentally. In particular, the band gap of semiconductors may be determined experimentally using optical spectroscopy. These values may be more relevant then the calculation from “geometrical optimization”.

We thank the reviewer for the comment. Ultraviolet-visible spectroscopy (UV-VIS) was conducted on both powders to compare experimental results with computational data. Due to a high reflection in comparison to a diffusion scattering that SLTO creates, the data could not be fitted appropriately. Therefore, the obtained band gap of about 3.50 eV possesses a pronounced uncertainty.

On the other hand, the experimentally obtained band gap for SCNPTO was defined to be 3.39 eV. Here we could observe similar differences between the experimental and calculated values for SLTO (0.45 eV) and SCNPTO (0.70 eV), respectively. The observed discrepancy is as expected and matches with a trend in the literature. We added this information in the Manuscript (Lines 158-162 and 320-327), and the data are displayed in Supplementary materials, Table S2 (Line 472 in the main text).

The following minor issues should be also corrected/commented:

  1. Figure1 –reference to the PDF card for SCNTPO should be added.

We agree with the Reviewer that it would be good to include a PDF of SCNTPO, but unfortunately, it is impossible since the new material is not present in the databases.

  1. Line85 and below – abbreviations PSD/PSA should be defined.

We thank the Reviewer for spotting this error. PSD is a misprint and has been corrected to PSA, the acronym for Particle Size Analysis. The changes are marked in the text (Line 111). We also added full names for the rest of the acronyms in lines 109-111.

  1. Line212 and Figure 3 – the mean diameter of the small particles equals to 200nm but not 20nm?

We thank the Reviewer for spotting this error; 200 nm is the right value. The misprint has been corrected (Lines 257 and 260).

  1. Figure4 – bars and lines for the different phases are hard to compare with the each other.

We thank the Reviewer for this suggestion. The values of both materials are now displayed as bars.

  1. Wavelength characteristics of the used light sources should be moved to the main text because spectral issues of photocatalytic process are discussed.

We thank the Reviewer for this suggestion. We have included the information in the manuscript (Lines 173-179).

  1. As stated in section 2.4 - photocatalytic activity was measured at the pH 7,5. However, pH changes in Figure 6 starts from the pH values about 9,5.

Thank you very much for noticing this; 7.50 is a misprint. We corrected it to 9.50 (Line 168).

  1. Figure 5d belongs to the SLTO phase?

We thank the Reviewer for spotting this mistake, SLTO is correct, and we have changed it accordingly (Line 318).

To sum up, the manuscript may be subjected to publication in Nanomaterials after careful revision according to the points listed above.

The authors thank the Reviewer very much for the valuable comments and positive opinion of our results. We hope the manuscript now meets the criteria for publication in Nanomaterials Journal.

Reviewer 2 Report

This work describes fabrication of SLTO and SCNPTO perovskites for photocatalytic activities. The work idea is good but the way they have performed this work is very weak. This should be revised extensively to reach any decision. 

1. The introduction part is very weak and they have mentioned that these materials were not tested before for this application, why? what was the limitations that these materials have not been tested?

2. Also, they should provide a comparison of this materials to the other perovskites then build a case for significance of this work.

3. The SEM of the nanoparticles is not very clear and the size of 30 nm is ambiguous too. They should calculate crystal size from XRD and relate to SEM size such as done in this work (https://doi.org/10.1016/j.chemosphere.2021.132525)

4. The photocatalytic responses are not discussed in details, what are the species that are actually responsible for degradation? they should perform more experiments or atleast cite some relevent literature which have done these kind of studies. 

5. Cyclic test are also missing in this work. refere to this work (https://doi.org/10.1016/j.jcis.2017.05.057)

6. A comparison of their materials performance with other oxide materials such as CeO2, PrO2 and few others in the form of a table will also be helpful.  some can be found here; https://doi.org/10.1016/j.ceramint.2021.02.117

7. The morphology after the cyclic test or atleast after degradation test should be provided.  

Author Response

Journal: Nanomaterials (ISSN 2079-4991)

Manuscript ID: nanomaterials-2017918

Title: Evaluation of photocatalytic performance of Sr0.9La0.1TiO3 and Sr0.25Ca0.25Na0.25Pr0.25TiO3 nano-sized ceramic powders for water purification

Section: Energy and Catalysis

Special Issue: Nanoscale Materials for Water Purification and Catalysis

The authors thank the reviewer for their constructive comments and recommendations. We have taken the comments on board to improve and clarify the Manuscript. Please find below a detailed point-by-point response to all comments.

The reviewers’ comments are typed in black. The responses to reviewers are tagged blue. The changes in the text of the Manuscript are typed in red. The line numbers in responses to reviewers refer to the corrected text.

Response Reviewer 2:

This work describes fabrication of SLTO and SCNPTO perovskites for photocatalytic activities. The work idea is good but the way they have performed this work is very weak. This should be revised extensively to reach any decision.

We thank the reviewer for the suggestions and the opportunity to improve our manuscript.

  1. The introduction part is very weak and they have mentioned that these materials were not tested before for this application, why? what was the limitations that these materials have not been tested?

Thank you very much for this comment. In the revised version of the manuscript, we have made efforts to improve the Introduction regarding Sr0.9La0.1TiO3 (SLTO) and Sr0.25Ca0.25Na0.25Pr0.25TiO3 (SCNPTO) materials application and their advances and limitations (Lines 66-70, 78-94, 101-106, 109-112). Also, we added new References 12 (Lines 527) and 14-24 (Lines 531-557).

  1. Also, they should provide a comparison of this materials to the other perovskites then build a case for significance of this work.

In the revised version of the Manuscript, we have added in the Introduction some previous results of modifications of perovskites and built a case of the significance of the Manuscript (Lines 78-94 and 101-106). We added new References 14-24 (Lines  531-557).

  1. The SEM of the nanoparticles is not very clear and the size of 30 nm is ambiguous too. They should calculate crystal size from XRD and relate to SEM size such as done in this work (https://doi.org/10.1016/j.chemosphere.2021.132525)

We thank the reviewer for the comment. Crystallite  sizes (CS) were obtained by HighScore Plus (Philips’ program) as estimated integral values for each peak position. The average CS of 27 and 33 nm were calculated for SLTO and SCNPTO, respectively, and the data is graphically presented in Figure S2. A more detailed analysis would require Rietveld refinement and clarification of the Instrument Resolution File (IRF), which would probably result in slightly different values than those given above. Since CS is always smaller than particle size, the calculated CS of SLTO (27 nm) and SCNPTO (33 nm) are sufficient to conclude that the particle size must be bigger than 30 nm, as initially stated . One can relate the CS with particle size through a degree of agglomeration, which basically gives an average number of crystallites present in a particle. The ratio of the volume of a particle (basically the cube of the particle size) to the volume of the crystallite (again cube of the crystalline size) gives the degree of agglomeration. Hence, in order not to be ambiguous, we have re-stated the particle sizes to be of about 50 and 70 nm (Line 27) for SLTO and SCNPTO, respectively.

The changes are made in the text accordingly (Lines 242-246) and we added Figure S2 in Supplementary materials (Line 470), and the work from N. Al-Najar et al. [47] is referred in the Reference (Lines 605), as suggested by the reviewer.

  1. The photocatalytic responses are not discussed in details, what are the species that are actually responsible for degradation? they should perform more experiments or atleast cite some relevent literature which have done these kind of studies.

Thank you for your suggestion. The discussion about species responsible for degradation has been added to the manuscript in part 3.2. Degradation efficiency (Lines 284-294) and References 48-50 (Lines 608-614).

  1. and 7. Cyclic test are also missing in this work. refere to this work (https://doi.org/10.1016/j.jcis.2017.05.057)

Unfortunately, we did not investigate the cyclic test, but we completely understand the rewiever’s point. We will definitively have in mind cyclic tests in future studies. In this manuscript we at least emphasized the importance of the cyclic test in the Introduction (Lines 66-70) and cited work [12] (Reference, Line 527) that the reviewer suggested.

  1. A comparison of their materials performance with other oxide materials such as CeO2, PrO2 and few others in the form of a table will also be helpful. some can be found here; https://doi.org/10.1016/j.ceramint.2021.02.117

Thank you very much for the suggestion. We improved the quality of our manuscript and compared results with results in the literature (Lines 428-441). Also, we added adequate reference 50 (Lines 612) and reference 57 (Lines 629).

Round 2

Reviewer 1 Report

The revised manuscript can be published in Nanomaterials.

Reviewer 2 Report

The authors addressed most of the comments from this reviewer. However i still feel that there is a need of further improvement and it is always welcome to address issues in the forthcoming researches.